# Alzheimer's disease risk gene *BIN1* induces Tau-dependent network hyperexcitability

Yuliya Voskobiynyk[1†], Jonathan R Roth[1†], J Nicholas Cochran[1], Travis Rush[1], Nancy VN Carullo[2], Jacob S Mesina[1], Mohammad Waqas[1], Rachael M Vollmer[1], Jeremy J Day[2], Lori L McMahon[3], Erik D Roberson[1*]

[1]Center for Neurodegeneration and Experimental Therapeutics, Alzheimer's Disease Center, and Evelyn F. McKnight Brain Institute, Departments of Neurology and Neurobiology, University of Alabama at Birmingham, Birmingham, United States; [2]Department of Neurobiology, University of Alabama at Birmingham, Birmingham, United States; [3]Department of Cell, Developmental and Integrative Biology, University of Alabama at Birmingham, Birmingham, United States

**Abstract** Genome-wide association studies identified the *BIN1* locus as a leading modulator of genetic risk in Alzheimer's disease (AD). One limitation in understanding *BIN1*'s contribution to AD is its unknown function in the brain. AD-associated *BIN1* variants are generally noncoding and likely change expression. Here, we determined the effects of increasing expression of the major neuronal isoform of human BIN1 in cultured rat hippocampal neurons. Higher BIN1 induced network hyperexcitability on multielectrode arrays, increased frequency of synaptic transmission, and elevated calcium transients, indicating that increasing BIN1 drives greater neuronal activity. In exploring the mechanism of these effects on neuronal physiology, we found that BIN1 interacted with L-type voltage-gated calcium channels (LVGCCs) and that BIN1–LVGCC interactions were modulated by Tau in rat hippocampal neurons and mouse brain. Finally, Tau reduction prevented BIN1-induced network hyperexcitability. These data shed light on BIN1's neuronal function and suggest that it may contribute to Tau-dependent hyperexcitability in AD.

*For correspondence:
eroberson@uabmc.edu

[†]These authors contributed equally to this work

## Introduction

Genetic discoveries have provided critical insights into potential mechanisms of Alzheimer's disease (AD), the most common neurodegenerative disease. Mutations in *APP*, *PSEN1*, or *PSEN2* cause early-onset, autosomal dominantly inherited AD, but are quite rare. Several more common genetic variants that increase AD risk to differing degrees have been identified. Among these, variants near *BIN1* have particularly high population attributable risk, because the risk allele is highly prevalent (~40% allele frequency for the index SNP, rs6733839) and has a relatively large effect size (odds ratio: 1.20; 95% confidence interval: 1.17–1.23) (*Kunkle et al., 2019*).

*BIN1* was first linked to AD in early genome-wide associated studies (GWAS) (*Harold et al., 2009*; *Seshadri et al., 2010*) and remains second only to *APOE* in genome-wide significance in the recent meta-analysis of 94,437 individuals by the International Genomics of Alzheimer's Disease Project (*Kunkle et al., 2019*). This association has been replicated in datasets with subjects from diverse genetic backgrounds (*Carrasquillo et al., 2011*; *Hollingworth et al., 2011*; *Hu et al., 2011*; *Lambert et al., 2011*; *Lee et al., 2011*; *Logue, 2011*; *Naj et al., 2011*; *Wijsman et al., 2011*; *Kamboh et al., 2012*; *Chapuis et al., 2013*; *Lambert et al., 2013*; *Liu et al., 2013*; *Miyashita et al., 2013*; *Reitz et al., 2013*; *Li et al., 2015*; *Dong et al., 2016*; *Rezazadeh et al., 2016*; *Wang et al., 2016*). Further, unbiased epigenetic analyses have provided independent evidence linking *BIN1* to

AD pathogenesis in several epigenome-wide association studies examining DNA methylation patterns in brain tissue from AD patients, in which *BIN1* again emerged as a top hit (*De Jager et al., 2014*; *Chibnik et al., 2015*; *Yu et al., 2015*). This association was also observed in tissue from preclinical AD patients, indicating that changes in *BIN1* methylation occur early in disease (*De Jager et al., 2014*; *Chibnik et al., 2015*). Also, associations between *BIN1* methylation and AD are independent of genetic variants identified in GWAS, providing an orthogonal line of evidence for BIN1's involvement in AD. Importantly, *BIN1* variants have been linked to earlier age of onset (*Naj et al., 2014*). In addition to GWAS reports examining polymorphisms associated with AD diagnosis by clinical criteria, other studies have examined genetic risk factors for AD neuropathology. BIN1 was significantly associated with both amyloid plaque and neurofibrillary tangle pathologies, strengthening the association with AD (*Beecham et al., 2014*). While these unbiased screens have convincingly implicated *BIN1* in AD pathogenesis, the mechanisms underlying the association are not yet known, and many important questions about how *BIN1* contributes to AD remain.

One of the main limitations is an incomplete understanding of BIN1's normal function in the brain. Its structure suggests that a key role may involve protein trafficking at the membrane, since all BIN1 isoforms contain an N-terminal BAR (BIN1/Amphiphysin/RVS167) domain that mediates membrane binding and curvature, plus a C-terminal SH3 domain that mediates protein–protein interactions, including with Tau (*Chapuis et al., 2013*; *Sottejeau et al., 2015*). The larger, neuron-specific isoforms also contain a clathrin-AP2 binding (CLAP) domain likely involved in endocytosis (*De Rossi et al., 2016*).

The *BIN1* variants associated with AD do not alter the coding sequence of BIN1 but are rather concentrated in a presumed regulatory region upstream of the promoter. Although BIN1 is ubiquitously expressed throughout the body, levels are highest in the brain and muscle (*Butler et al., 1997*), and its most critical role is in the heart, as homozygous deletion of murine *Bin1* causes early lethality due to severe ventricular cardiomyopathy (*Muller et al., 2003*). Canonically, BIN1 plays a role in protein trafficking and endocytosis, specifically trafficking L-type voltage gated calcium channels (LVGCCs) in cardiac myocytes to the membrane to strengthen calcium signaling (*Hong et al., 2010*). However, the function BIN1 plays in neurons remains much less clear.

In this study, we addressed BIN1's role in neurons by expressing the predominant neuronal BIN1 isoform (isoform 1) in primary hippocampal neuron cultures. Our studies revealed a role for BIN1 in regulating neuronal activity and a potential molecular mechanism involving its interactions with calcium channel subunits.

## Results

### Higher BIN1 induces network hyperexcitability

To begin studying the effects of altered BIN1 levels in neurons, we first used AAV to express the predominant neuronal isoform of human BIN in primary rat hippocampal neuronal cultures. We verified expression of BIN1 using an mKate2 fluorophore fused to the C-terminus. A construct encoding mKate2 alone was used as a control. We determined that BIN1 expression increased ~8–9-fold by immunocytochemistry and remained stable up to 3.5 weeks post transduction (*Figure 1A–C*). Higher BIN1 did not change neuronal morphology (*Figure 1B*), the total number of neurons per well (*Figure 1D–E*), nor the resting membrane potential (RMP) or input resistance ($R_{in}$) of cultured neurons (*Table 1*), indicating no significant toxic or trophic effect of overexpressing BIN1 under these conditions.

We then recorded action potentials and burst firing in these neurons on multielectrode arrays (MEAs) after 10 days (*Figure 1F–G*). Local field potential (LFP) traces representing neuronal action potential and burst firing were recorded for 20 min then analyzed (*Figure 1H–I*). We found that higher BIN1 levels were associated with increased frequency of action potentials (2.3-fold, *Figure 1J*) and action potential bursts (2.1-fold, *Figure 1K*). There was no change in the total number of active neurons on MEAs (*Figure 1L*).

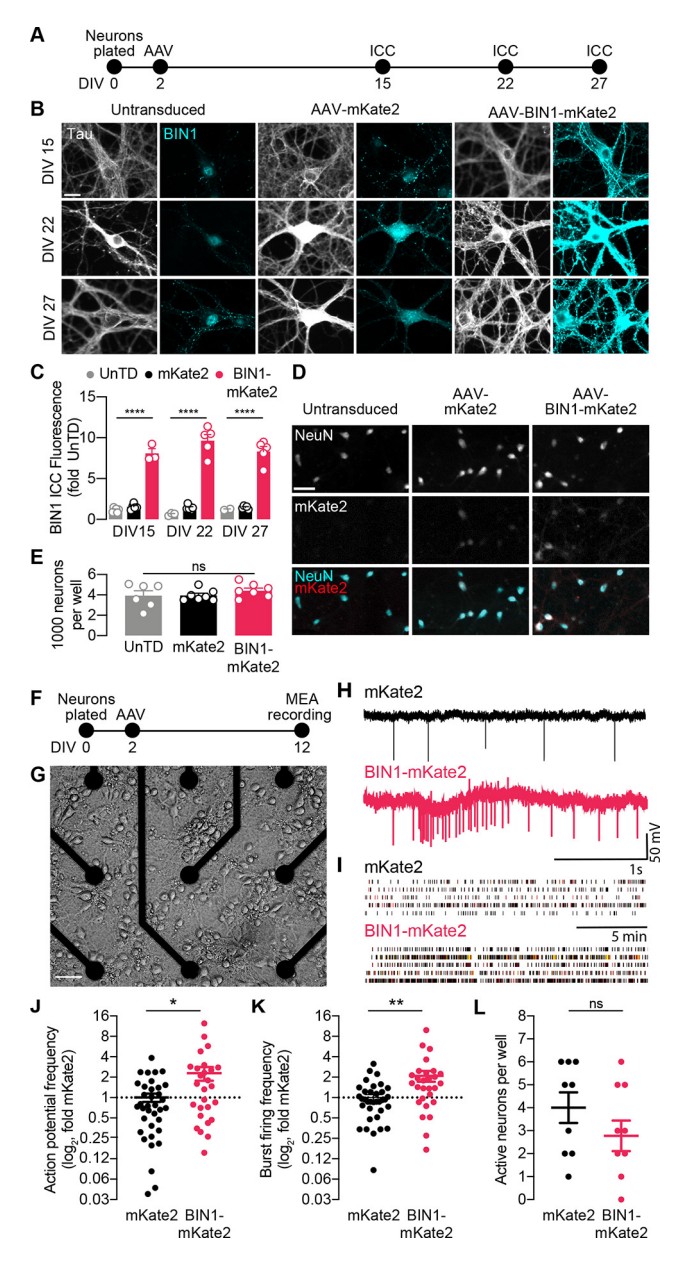

**Figure 1.** BIN1 increases action potential and burst frequency in primary hippocampal neurons cultured on microelectrode arrays (MEAs). (**A**) ICC experimental timeline: neurons were plated on day in vitro (DIV) 0, virally transduced on DIV 2, and immunostained at DIV 15, DIV 22, or DIV 27. (**B**) Representative images of primary hippocampal cultures: untransduced (left), AAV-mKate2 (center), or AAV-BIN1-mKate2 (right), showing Tau and BIN1 immunostaining at DIV 15 (top), DIV 22 (middle), or DIV 27 (bottom). Scale bar = 20 μm. (**C**) AAV-BIN1-mKate2 increased BIN1 levels ~ 8–9-fold in BIN1 group compared to BIN1 levels in untransduced or mKate2 groups (*n* = 2–6 fields of view per coverslip, 60x magnification, two-way ANOVA, BIN-DIV interaction p=0.1123, main effect of AAV-BIN1-mKate2 ****p<0.0001, main effect of DIV p=0.6373, Tukey's multiple comparisons test: DIV 15:UnTD vs. DIV 15:AAV-BIN1-mKate2, ****p<0.0001, DIV 22:UnTD vs. DIV 22:AAV-BIN1-mKate2, ****p<0.0001, DIV 27:UnTD vs. DIV 27:AAV-BIN1-mKate2, ****p<0.0001. (**D**) Representative images of primary hippocampal cultures at DIV 12: untransduced (left), AAV-mKate2 (center), or AAV-BIN1-mKate2 (right), showing NeuN immunostaining (top), mKate2 fluorescence (middle), or merge of both (bottom). Scale bar = 25 μm. (**E**) The total number of neurons per well did not change between untransduced, mKate2, or BIN1 groups (*n* = 6–7 coverslips, 10 × 10 fields of view per coverslip, 20x magnification, from two different primary neuron harvests, one-way ANOVA, p=0.5157). (**F**) MEA experimental timeline: neurons were plated on day in vitro (DIV) 0, virally

*Figure 1 continued on next page*

*Figure 1 continued*

transduced on DIV 2, and recorded on DIV 12. (G) Primary neuronal hippocampal cultures grown on an MEA plate. Scale bar = 50 μm. (H) Representative local field potential (LFP) traces. (I) Representative raster plots of firing activity from five different neurons per group. (J) BIN1 increased action potential frequency (*n* = 27–36 neurons per group from three different primary neuron harvests, normalized to the controls from each harvest, median frequency in controls = 388 mHz; unpaired Mann-Whitney U test; p=0.0233). (K) BIN1 increased burst firing frequency (*n* = 27–36 neurons per group from three different primary neuron harvests, normalized to the controls from each harvest, median frequency in controls = 11.7 mHz; unpaired Mann-Whitney U test; p=0.0020). (L) The total number of active neurons per well did not differ between mKate2 and BIN1 expressing groups (*n* = 9 MEA plates for each group from three different primary neuron harvests, unpaired Student's t test; p=0.346). All data are expressed as mean ± SEM, *p<0.05, **p<0.01, and ****p<0.0001. All data are expressed as mean ± SEM.

## Higher BIN1 increases frequency of spontaneous excitatory and inhibitory synaptic transmission

Since higher BIN1 levels increased action potential and burst frequencies in the MEA recordings, we hypothesized that this would be associated with an increased frequency of spontaneous excitatory postsynaptic currents (sEPSCs). To test this, we used whole-cell voltage-clamp recordings from BIN1-transduced neurons at DIV 19–21 (*Figure 2A*), pharmacologically isolating sEPSCs using picrotoxin to block inhibitory GABA$_A$R currents. (*Figure 2B*). Consistent with the increased action potential frequency observed in MEA recordings (*Figure 1E*), higher BIN1 levels were associated with dramatically increased sEPSC frequency (interevent interval decreased >50%) (*Figure 2C*). sEPSC amplitudes differed by <10% (*Figure 2D*).

To investigate if this effect of higher BIN1 levels was selective for excitatory transmission, we next examined whether higher BIN1 expression had similar effects on spontaneous inhibitory postsynaptic currents (sIPSCs). We determined the proportion of GABAergic interneurons in our primary hippocampal cultures and found that 10% of the neurons were GAD67 positive (*Figure 2E*), consistent with prior work (*Benson et al., 1994*). To determine the effect of higher BIN1 levels on inhibitory synaptic transmission from these neurons, we recorded pharmacologically isolated GABA$_A$R-mediated sIPSCs using DNQX, APV, and nifedipine to block AMPARs, NMDARs, and L-type voltage-gated calcium channels (LVGCCs), respectively (*Figure 2F*). Similar to the effect on sEPSC frequency, higher BIN1 increased sIPSC frequency (decrease in interevent interval, *Figure 2G*). There was a coincident decrease in sIPSC amplitude (*Figure 2H*).

Overall, these findings suggest that higher BIN1 levels increase the frequency of both sEPSCs and sIPSCs in primary hippocampal cultures, agreeing with the increased action potential firing observed using MEAs (*Figure 1E–F*).

## Higher BIN1 in mature neurons increases calcium influx

Using AAVs requires transduction soon after plating (DIV 2) because of the time required for transgene expression, so some effects could be due to increasing BIN1 levels during early neuronal development. To dissociate the effect of higher BIN1 on network hyperexcitability from neuronal development, we transiently transfected primary hippocampal cultures at DIV 14, when neurons are more fully developed (*Figure 3A*). We co-transfected BIN1 constructs with the genetically encoded calcium indicator, GCaMP6f, which allows for single neuron calcium imaging in primary hippocampal cultures. We used two BIN1 constructs, both based on human isoform one tagged with the mKate2

**Table 1.** Resting membrane potential (RMP) and input resistance (R$_{in}$) in patched hippocampal neurons did not differ across untransduced, AAV-mKate2, and AAV-BIN1-mKate2 groups.

|  | RMP, mV | R$_{in}$, MΩ | N, Cells |
|---|---|---|---|
| Untransduced | –60.43 ± 5.36 | 843.04 ± 55.11 | 4 |
| AAV-mKate2 | –59.90 ± 3.70 | 834.83 ± 41.09 | 6 |
| AAV-BIN1-mKate2 | –62.40 ± 2.24 | 790.18 ± 15.74 | 7 |
| One-way ANOVA, *p* | 0.85 | 0.36 |  |

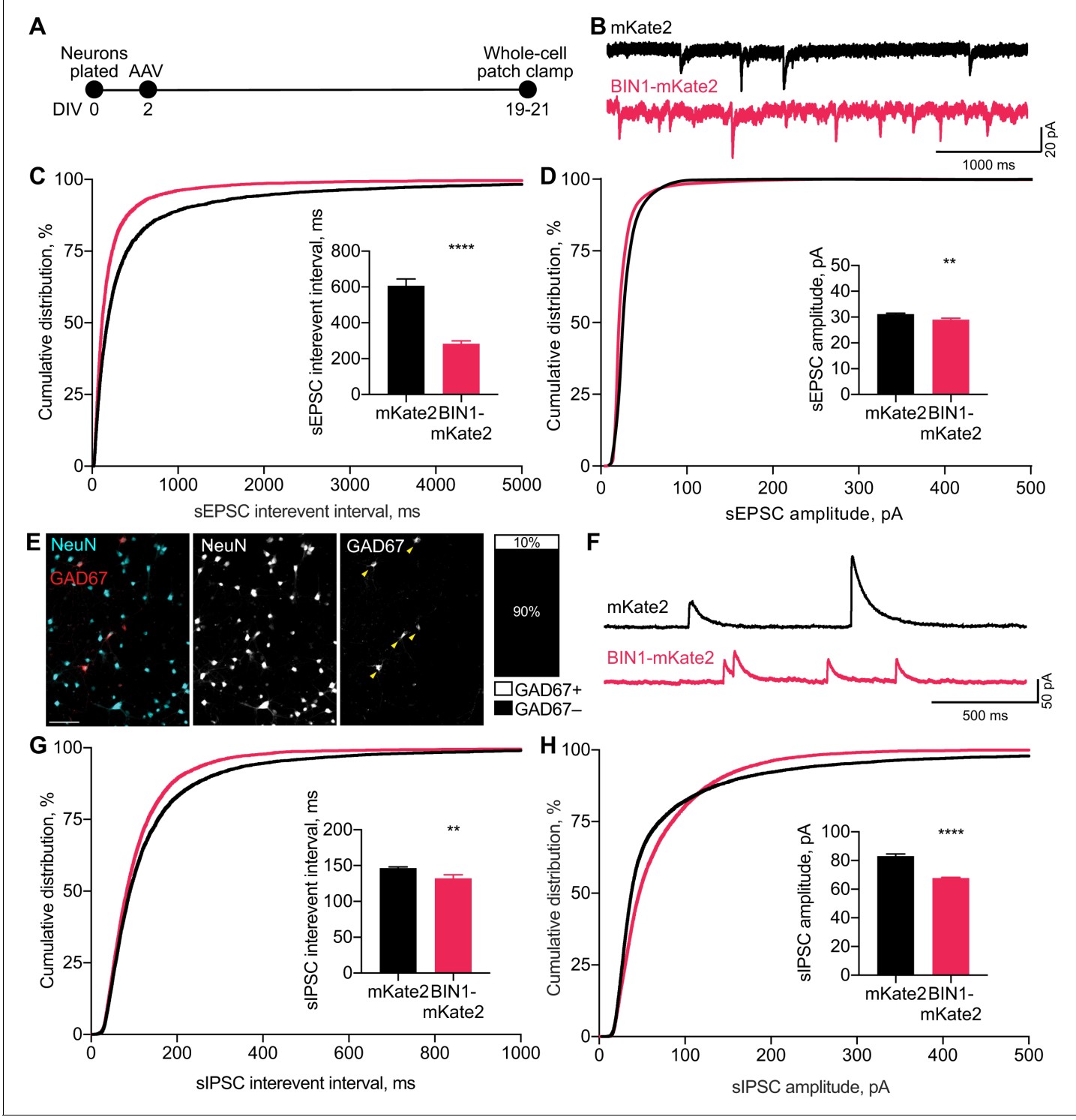

**Figure 2.** BIN1 increases both excitatory and inhibitory synaptic transmission. (**A**) Synaptic transmission recordings experimental timeline: neurons were plated on DIV 0, virally transduced on DIV 2, electrophysiologically recorded on DIV 19–21. (**B**) Representative traces of sEPSCs recorded from neurons transduced with mKate2 or BIN1. (**C**) BIN1 decreased mean sEPSC interevent interval (Kolmogorov-Smirnov test on cumulative distribution, ****p<0.0001, KS D score: 0.1657; unpaired two-tailed Student's t-test on mean IEI, ****p<0.0001). (**D**) BIN1 slightly decreased mean sEPSC amplitude unpaired (Kolmogorov-Smirnov test on cumulative distribution, ****p<0.0001, KS D score: 0.1803; unpaired two-tailed Student's t-test on mean amplitude, ***p=0.0004) (*n* = 12–21 neurons per group from three different primary neuron harvests). (**E**) Representative images and quantification of NeuN and GAD67+ neurons in primary hippocampal cultures at DIV 12 (*n* = 255 GAD67+ neurons, *n* = 2342 NeuN+ neurons, from 10 randomly taken images per coverslip, 10 coverslips from two different primary neuron harvests.) Scale bar = 100 µm. (**F**) Representative traces of sIPSCs recorded from

*Figure 2 continued on next page*

*Figure 2 continued*

neurons transduced with mKate2 or BIN1. (G) BIN1 decreased mean sIPSC interevent interval (Kolmogorov-Smirnov test on cumulative distribution, ****p<0.0001, KS D score: 0.06862; unpaired two-tailed Student's t-test on mean IEI, **p=0.0035) (H) BIN1 decreased mean sIPSC amplitude (Kolmogorov-Smirnov test on cumulative distribution, ****p<0.0001, KS D score: 0.1297; unpaired two-tailed Student's t-test on mean amplitude, ****p<0.0001) ($n$ = 11–16 neurons per group from three different primary neuron harvests). All data are expressed as mean ± SEM.

fluorophore. In addition to the full-length BIN1 construct used in *Figures 1–2*, we also used a construct engineered to remove the BAR domain (ΔBAR), which is predicted to abolish BIN1 membrane localization, and thus likely its activity. As before, mKate2 alone was used as a control construct. Interestingly, the pattern of mKate2 distribution within the neurons was strikingly different across groups, as mKate2 and BIN1-ΔBAR exhibited diffuse localization throughout the soma, neurites, and nucleus, while wild-type BIN1 had a punctate distribution throughout the cytosol but was excluded from the nucleus (*Figure 3B*). These observations agree with BAR-domain dependent membrane localization of BIN1 found in other cell types (*Hong et al., 2010*; *Picas et al., 2014*).

We monitored basal calcium activity of individual transfected neurons by imaging GCaMP fluorescence using laser scanning microscopy. We measured the change in somatic GCaMP fluorescence intensity relative to the quiescent period between transients (defined as $F_0$) and classified neurons as either active ($\geq$1 calcium transient) or inactive (no calcium transients) (*Figure 3C*).

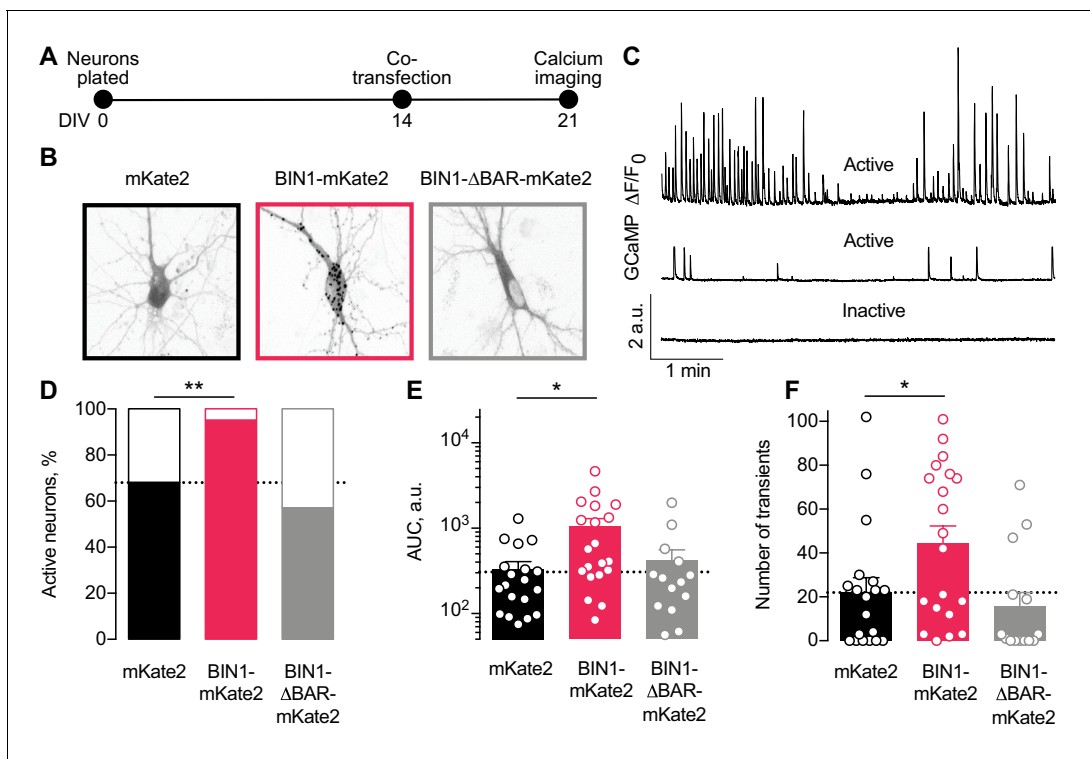

**Figure 3.** Higher BIN1 levels in mature neurons increase calcium influx in primary hippocampal neuronal cultures. (A) Calcium imaging experimental timeline: neurons were plated on DIV 0, co-transfected on DIV 14 with GCaMP6f calcium indicator and either BIN1-mKate2, BIN1-ΔBAR-mKate2, or mKate2 control construct, and recorded on DIV 21. $n$ = 14–20 neurons per condition. (B) mKate2 fluorescence in primary transfected primary hippocampal neurons. BIN1-mKate2 fluorescence was punctate in transfected neurons. mKate2 and BIN-BAR-mKate2 fluorescence was diffuse and filled the neuron. (C) GCaMP fluorescence intensity, F, relative to the quiescent period between transients, $F_0$. Neurons were classified as either active (with a range of activity levels indicated by the top and middle traces) or inactive (no calcium transients during the 8 min recording, bottom trace). (D) BIN1, but not BIN1-ΔBAR, increased the proportion of active neurons (Binomial test, **p=0.0071). (E) BIN1, but not BIN1-ΔBAR, increased neuronal calcium influx measured as area under the curve (AUC; one-way ANOVA, p=0.0134; Dunnett's posthoc, *p=0.0122). (F) BIN1, but not BIN1-ΔBAR, increased the number of calcium transients (one-way ANOVA, p=0.0144; Dunnett's posthoc, mKate2 vs BIN1-mKate2 adjusted *p=0.0437). All data are expressed as mean ± SEM.

About a third of neurons were inactive under control conditions (*Figure 3D*), consistent with prior studies (*Kuijlaars et al., 2016*; *Lerdkrai et al., 2018*). However, neurons expressing the full-length human BIN1 construct were almost never inactive (*Figure 3D*). As expected, the BIN1-ΔBAR construct was similar to controls (*Figure 3D*), indicating the importance of BIN1 membrane localization for the effect. Increasing BIN1 levels roughly doubled calcium influx as measured by both area under the curve (*Figure 3E*) and the number of calcium transients (*Figure 3F*).

## BIN1 interacts with LVGCCs in neurons

We were interested to find that the ability of BIN1 to increase neuronal activity was dependent on the presence of the BAR domain, which is critical for its membrane localization. One of BIN1's known functions outside of the brain is localizing LVGCCs to the membrane of cardiomyocyte T-tubules (*Hong et al., 2010*; *Hong et al., 2014*). Therefore, we asked if BIN1 interacts with LVGCCs in neurons and if increased interaction between BIN1 and LVGCCs could be a potential mechanism by which BIN1 increases neuronal activity.

To begin addressing this question, we first examined BIN1 interactions with LVGCC beta-1 subunits (LVGCC-β1), which reside on the inner face of the membrane and target LVGCCs to the membrane (*Buraei and Yang, 2010*). To determine whether BIN1 and LVGCC-β1 interact directly, we used proximity ligation assay (PLA), which allows quantification and visualization of protein-protein interactions in situ, producing a fluorescent punctum whenever the two antibody epitopes are within 40 nm (i.e., directly interacting or nearby in a complex). We detected endogenous BIN1–LVGCC-β1 interaction in neuronal soma and neurites of untransduced neurons (*Figure 4A*). If a BIN1-mediated effect on LVGCC's underlies the observed effects on neuronal activity, then AAV-BIN1 constructs should increase the interaction (*Figure 4B*). Transduction with BIN1-mKate2 substantially increased BIN1–LVGCC-β1 interaction, while transduction with the mKate2 control vector did not change endogenous interaction levels (*Figure 4C,D*).

## BIN1-LVGCC interaction is Tau-dependent

Multiple studies have demonstrated that BIN1 directly interacts with Tau, both in vitro and in vivo (*Chapuis et al., 2013*; *Lasorsa et al., 2018*; *Sartori et al., 2019*). This interaction between BIN1 and Tau is mediated by the SH3 domain of BIN1 and PxxP motifs in Tau's central proline-rich region. Interestingly, LVGCC-β1 also harbors an SH3 domain that could also interact with Tau. Thus, we hypothesized that the BIN1 interaction with LVGCC-β1 might be at least in part scaffolded by Tau (*Figure 5A*).

We first used a live-cell bioluminescence resonance energy transfer (BRET) assay (*Cochran et al., 2014*) to determine if Tau interacts with the SH3 domains of both BIN1 and LVGCC-β1. We transfected CHO cells with Tau-mKate2 (acceptor) and either the BIN1 SH3 domain or LVGCC-β1 SH3 domain tagged with click beetle green (donor) (*Figure 5B*). Both Tau–BIN1 and Tau–LVGCC-β1 demonstrated BRET, indicating that Tau interacts with each of these SH3 domains (*Figure 5C*).

We then tested the hypothesis that Tau affects the BIN1–LVGCCβ1 interaction, using the BIN1–LVGCC-β1 PLA assay with and without pretreatment with Tau antisense oligonucleotide (ASO) (*Figure 5D*). We recently demonstrated that this ASO reduces Tau protein by about 50% under these conditions (*Rush et al., 2020*). Tau reduction decreased BIN1–LVGCC-β1 interaction in primary hippocampal neurons, compared to neurons treated with a scrambled control ASO (*Figure 5E*), indicating that in cultured neurons the BIN1–LVGCC-β1 interaction is partially Tau-dependent.

Next, we determined if the BIN1–LVGCC-β1 interaction is also Tau-dependent in vivo. Using cortical brain lysates from wild-type and Tau knockout (Tau KO) mice, we immunoprecipitated LVGCC-β1 and blotted for BIN1. BIN1 co-immunoprecipitated with LVGCC-β1 from these brain lysates, and the BIN1–LVGCC-β1 complex was reduced in Tau KO brains, without any difference in LVGCC-β1 immunoprecipitation (*Figure 5F*). Taken together, these data indicate that the BIN1–LVGCC interaction is partially Tau-dependent both in vitro and in vivo.

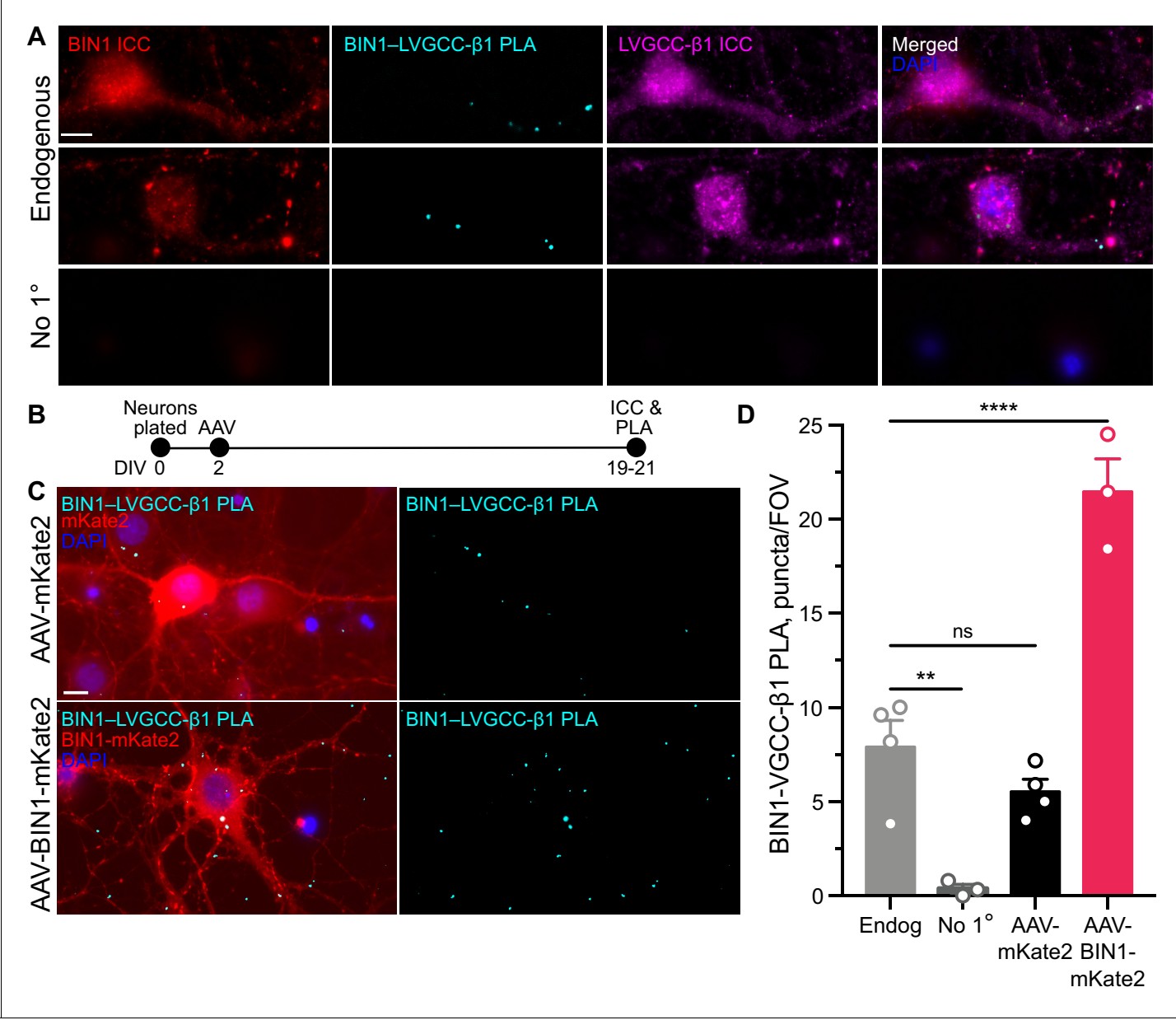

**Figure 4.** BIN1 interacts with LVGCC-β1 subunits in neurons. (**A**) Localization of endogenous BIN1, LVGCC-β1, and endogenous BIN1-LVGCC-β1 interaction detected by PLA. Scale bar = 10 µm. (**B**) Experimental timeline: neurons were plated on DIV 0, transduced with AAV-BIN1-mKate2 or AAV-mKate2 on DIV 2, and fixed and stained on DIV 19–21. (**C**) Representative images of mKate2 fluorescence, BIN1–LVGCC-β1 PLA puncta, and BIN1 ICC in primary hippocampal neurons. Scale bar = 10 µm. (**D**) BIN1–LVGCC-β1 interaction was increased by BIN1 (*n* = 3–4 coverslips per group, each with 5 fields of view averaged, from three different primary neurons harvests; one-way ANOVA, p<0.0001; Endogenous vs. AAV-BIN1-mKate2 ****p=0.0001 by Dunnett's post-hoc). All data are expressed as mean ± SEM.

## Tau reduction prevents network hyperexcitability induced by higher BIN1

Tau reduction is protective in many models of AD and it reduces network hyperexcitability in many disease models, including AD and epilepsy models (*Chin et al., 2007*; *Roberson et al., 2007*; *Holth et al., 2011*; *Roberson et al., 2011*; *DeVos et al., 2013*; *Gheyara et al., 2014*; *Liu et al., 2017*). Thus, since Tau reduction decreases BIN1–LVGCC-β1 interaction in primary hippocampal neurons and brain homogenates, we asked whether Tau reduction attenuates network hyperexcitability induced by increased BIN1.

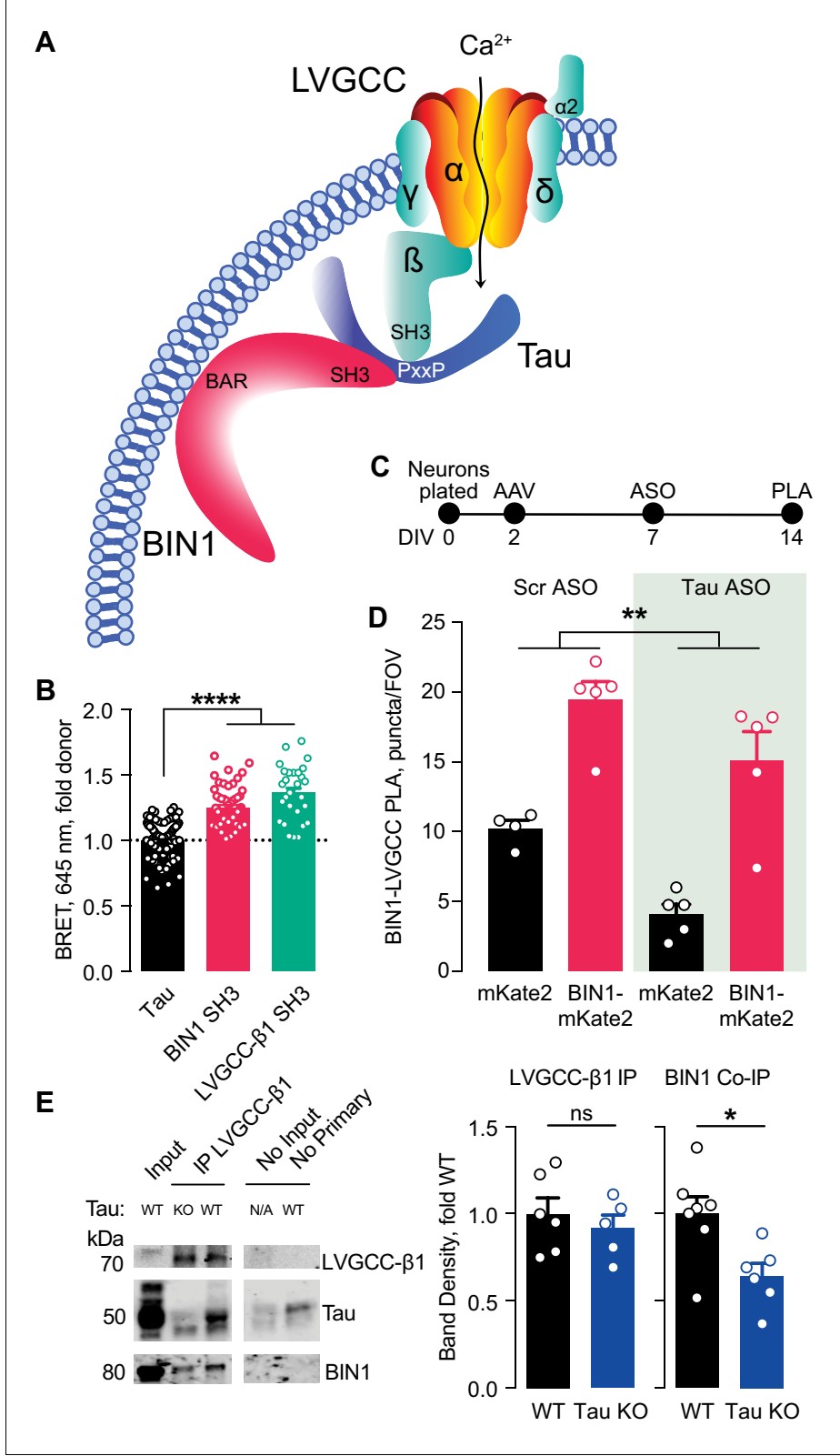

**Figure 5.** Tau-dependent BIN1–LVGCC interaction. (**A**) Model of Tau-dependent BIN1–LVGCC interaction. BIN1's BAR domain localizes BIN1 to the plasma membrane, and PxxP motifs in Tau's central proline-rich domain interact with the SH3 domains of BIN1 and LVGCC- β1. (**B**) Tau interacts with both BIN1 and LVGCC-β1 SH3 domains detected by bioluminescence resonance energy transfer (BRET) (*n* = 28–164 wells, one-way ANOVA,

*Figure 5 continued on next page*

*Figure 5 continued*

****p<0.0001). (**C**) Experimental timeline of BIN1–LVGCC-β1 PLA: neurons were plated on DIV 0, transduced with AAV-BIN1-mKate2 or AAV-mKate2 on DIV 2, treated with Tau or scrambled ASO on DIV 7, and stained on DIV 14. (**D**) AAV-BIN1-mKate2 increased BIN1-LVGCC-β1 interaction, while Tau reduction with Tau ASO decreased BIN1–LVGCC-β1 interactions (n = 4–6 coverslips per group representing an average of 4–5 fields of view (FOV) per coverslip from three different neuronal harvests; Two-way ANOVA, main effect of Tau ASO **p=0.0018, main effect of AAV-BIN1-mKate2 ****p<0.0001). (**E**) We immunoprecipitated LVGCC-β1 followed by western blotting for LVGCC-β1, Tau, and BIN1 from cortical homogenates of wild-type and Tau KO mice. The amount of LVGCC-β1 immunoprecipitated did not differ between wild-type and Tau KO brains (n = 5–6 mice, 3.56 ± 0.04 months old, unpaired Student's t test; p=0.5105). However, the amount of BIN1 co-immunoprecipitated with LVGCC-β1 was decreased in Tau KO brains compared to wild-type litter mate controls (n = 5–6 mice, unpaired Student's t test; *p=0.0157). All data are expressed as mean ± SEM.

To do this, we utilized a 48-well MEA system to permit recordings from many neurons with in-plate controls for each experiment. We grew neurons on the MEA, transduced them with AAV-BIN1 or AAV-mKate2 control, applied Tau ASO or a scrambled ASO control, then recorded neuronal activity (*Figure 6A*). As in our initial experiments, these manipulations did not affect the number of active neurons (*Figure 6B*), and higher BIN1 levels increased neuronal firing in this system as well (*Figure 6C–D*). Tau reduction completely blocked the BIN1-induced increases in action potential frequency (*Figure 6E–F*) and bursting (*Figure 6G–H*). These results demonstrate that BIN1-induced network hyperexcitability is Tau-dependent and add to the body of work demonstrating beneficial effects of Tau reduction on limiting network hyperexcitability and AD-related dysfunction.

## Discussion

Genetic data indicate that BIN1 can play an important role in AD pathogenesis, but a major limitation is the relatively poor understanding of BIN1's function in the central nervous system. We found that expressing the predominant human BIN1 isoform in primary hippocampal cultures led to a Tau-dependent increase in neuronal activity leading to network hyperexcitability. Higher BIN1 levels increased the frequency of both spikes and bursts recorded with multielectrode arrays (*Figure 1*). Using patch-clamp recordings of neurons overexpressing BIN1, we observed increased frequency of both excitatory and inhibitory synaptic transmission (*Figure 2*). Similarly, elevating BIN1 levels also increased calcium spikes in neurons co-transfected with the calcium indicator GCaMP6f (*Figure 3*). To understand the potential mechanism of increased calcium influx, we explored potential interactions with LVGCCs, which contribute to BIN1 effects on cardiac excitability. BIN1 interacted with LVGCCs in neurons in a Tau-dependent manner, assessed by both proximity ligation assay in cultured neurons and co-immunoprecipitation from brain (*Figures 4–5*). Finally, using a high-content multielectrode array system, we showed that Tau reduction prevented network hyperexcitability induced by BIN1 (*Figure 6*). Together, these data show Tau-dependent regulation of neuronal activity by the Alzheimer's disease risk gene *BIN1* and generate new insights about the mechanistic role BIN1 may play in AD.

Increasing evidence supports the idea that changes in neuronal excitability may contribute to AD pathogenesis. Functional imaging studies reveal hyperactivation of many brain regions in AD patients (*Dickerson et al., 2005*; *Hämäläinen et al., 2007*). This is an early event in AD pathogenesis, seen in asymptomatic individuals at genetic risk for AD (*Bookheimer et al., 2000*; *Reiman et al., 2012*). In addition, childhood epilepsy can drive subsequent amyloid accumulation (*Joutsa et al., 2017*). Furthermore, seizures are more frequent in AD patients than in age-matched controls (*Amatniek et al., 2006*; *Palop and Mucke, 2009*; *Scarmeas et al., 2009*), and in the early stages of disease, even patients without overt seizures often have epileptiform activity on neurophysiological recordings (*Vossel et al., 2013*; *Vossel et al., 2016*). Even more importantly, late-onset unprovoked seizures in older veterans are associated with a 2-fold risk of developing dementia, likely a first sign of neurodegenerative disease (*Keret et al., 2020*). Hyperexcitability is also seen in mouse models of AD, many of which have seizures (often nonconvulsive) and epileptiform spikes (*Palop et al., 2007*; *Minkeviciene et al., 2009*), excitation-inhibition imbalance in synaptic recordings (*Roberson et al., 2011*), and increased intrinsic neuronal excitability (*Brown et al., 2011*).

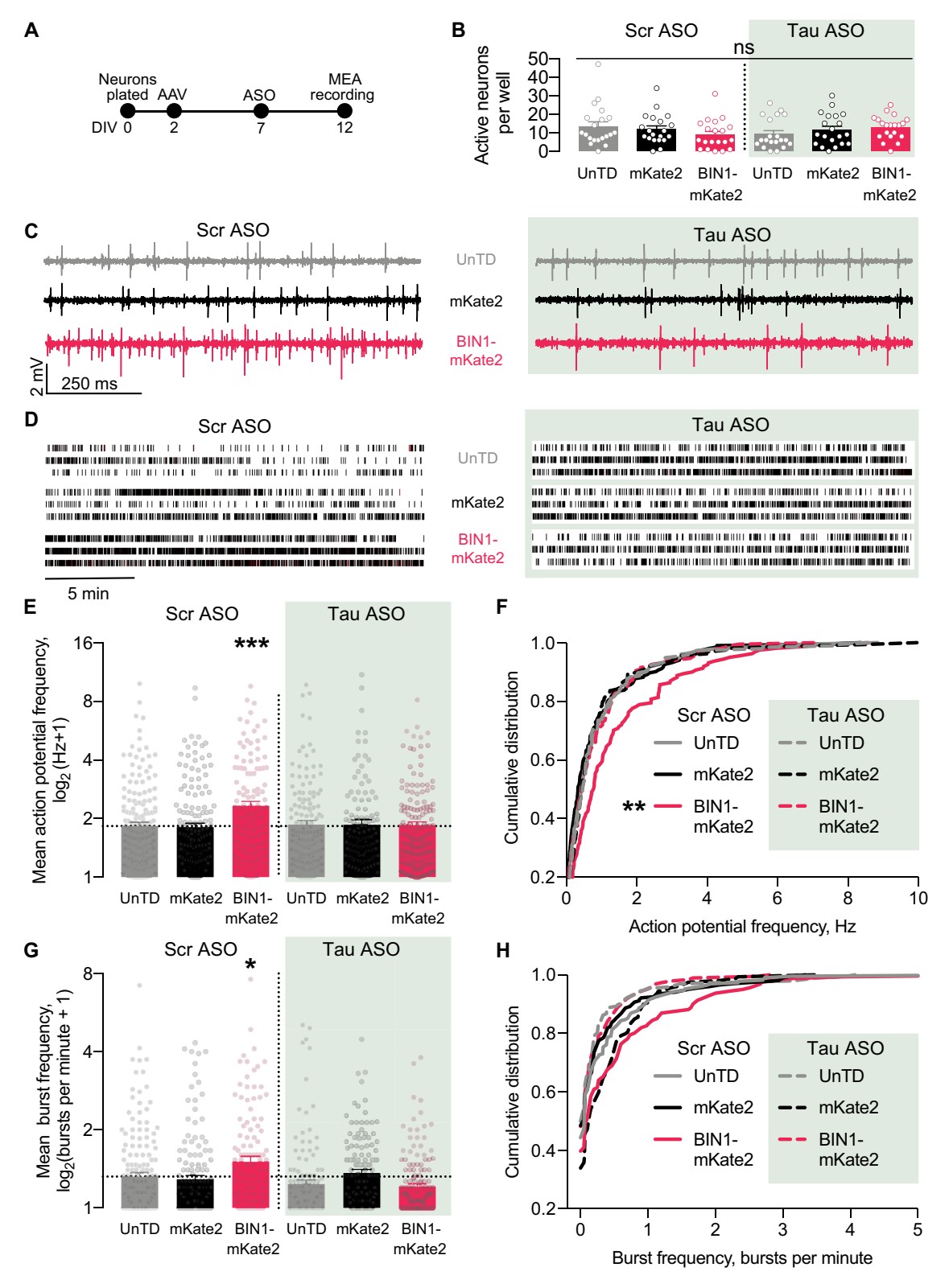

**Figure 6.** Tau reduction prevents network hyperexcitability induced by BIN1. (**A**) MEA experimental timeline: neurons were plated on DIV 0, virally transduced on DIV 2, treated with Tau or scrambled ASO on DIV 5, and electrophysiologically recorded on DIV 12. (**B**) The number of active neurons was not different between groups ($n$ = 6–8 coverslips per group from three different neuronal harvests; two-way ANOVA, main effect of Tau ASO p=0.9140, main effect of AAV-BIN1-mKate2 p=0.9026, interaction p=0.1101). (**C**) Representative LFP traces of MEA recordings. (**D**) Representative raster

*Figure 6 continued on next page*

*Figure 6 continued*

plots of MEA recordings. (**E**) Tau reduction prevented BIN1-induced network hyperexcitability as measured by mean action potential frequency (*n* = 159–230 neurons from 6 to 8 wells per group from three different neuronal harvests; two-way ANOVA, BIN-Tau interaction **p=0.0073, main effect of Tau ASO p=0.0761, main effect of AAV-BIN1-mKate2 *p=0.0130, Sidak's multiple comparisons test: UnTd:Scr ASO vs. BIN1-mKate2:Scr ASO ***p=0.0010). (**F**) Cumulative distribution of the mean action potential frequency (Kolmogorov-Smirnov test on cumulative distribution, UnTD-Scr ASO vs. BIN1-mKate2-Scr ASO **p=0.0028). (**G**) Tau reduction prevented BIN1-induced network hyperexcitability as measured by mean burst frequency (*n* = 159–230 neurons from 6 to 8 wells per group from three different neuronal harvests; two-way ANOVA, BIN-Tau interaction ***p=0.0005, main effect of Tau ASO **p=0.0066, main effect of AAV-BIN1-mKate2 p=0.2286, Sidak's multiple comparisons test: UnTd:Scr ASO vs. BIN1-mKate2:Scr ASO *p=0.0227). (**H**) Cumulative distribution of the burst frequency (Kolmogorov-Smirnov test on cumulative distribution, UnTD-Scr ASO vs. BIN1-mKate2-Scr ASO, p=0.1107). All data are expressed as mean ± SEM.

Existing data are consistent with a potential role for BIN1 in controlling neuronal excitability. As a membrane scaffolding protein, BIN1 promotes T-tubule formation in skeletal muscles (*Tjondrokoesoemo et al., 2011*). In cardiomyocytes, Bin1 traffics LVGCCs to T-tubules, allowing for proper T-tubule formation and excitation-contraction coupling (*Hong et al., 2010*; *Hong et al., 2014*). Genetic loss of Bin1 in cardiomyocytes decreases surface localization of LVGCCs to T-tubules and decreases LVGCC calcium transients (*Hong et al., 2010*). In addition, acute knockdown of Bin1 in primary cortical neurons reduced calcium spikes in response to NMDA (*McAvoy et al., 2019*). Similarly, a chronic genetic deletion of Bin1 in mice decreased mEPSCs frequency, suggesting an effect of Bin1 on synaptic transmission (*De Rossi et al., 2020*). Complementary to these studies on effects of decreased Bin1 expression in mice, we found that increased human BIN1 expression increases synaptic transmission, neuronal activity, and calcium transients, and that BIN1 interacts with LVGCCs in neurons. Moreover, we found that higher BIN1 increases not only excitatory, but also inhibitory synaptic transmission. Human genetics also support a link between BIN1 and network hyperexcitability, as the risk allele of the rs744373 variant upstream of *BIN1*, which is linked to AD, is also associated with impaired memory in temporal lobe epilepsy patients (*Bungenberg et al., 2016*).

Further studies will be needed to elucidate the precise mechanisms by which BIN1 regulates neuronal firing, but our studies suggest an effect on surface trafficking of LVGCCs. While biophysical and pharmacological properties of LVGCCs are tightly controlled by the principal $\alpha1$ subunit, the cytosolic auxiliary $\beta$ subunit plays an essential role in trafficking of LVGCCs to the plasma membrane (*Buraei and Yang, 2010*). Our study revealed BIN1 interaction with these LVGCC $\beta$ subunits in neurons (*Figures 4–5*), likely contributing to LVGCC neuronal surface localization. LVGCCs modulate neuronal firing (*Liu et al., 2014*) and control both basal and bursting neuronal activity through somatic and dendritic $Ca^{2+}$ transients (*Morton et al., 2013*; *Liu et al., 2014*). LVGCCs are also linked to neurodegeneration by carrying toxic amounts of $Ca^{2+}$ through an increase of LVGCC activity, density, or exposure to $\beta$-amyloid peptides (*Cataldi, 2013*).

Our findings suggest that the effects of BIN1 on neuronal excitability likely involve Tau. A variety of evidence has linked BIN1 to Tau in studies of AD. In AD patients, the *BIN1* risk variant, rs744373, is associated with increased Tau-PET levels, as well as reduced functional connectivity and impaired memory (*Zhang et al., 2015*; *Franzmeier et al., 2019*). The fact that BIN1 localizes in a complex with Tau (*Figures 4–5*; *Chapuis et al., 2013*; *Zhou et al., 2014*; *Sottejeau et al., 2015*; *Bretteville et al., 2017*; *Malki et al., 2017*; *Lasorsa et al., 2018*) supports the hypothesis that BIN1–Tau interaction regulates neuronal excitability, as there is now abundant data that a key function of Tau is regulating neuronal excitability, particularly susceptibility to hyperexcitability. This includes the fact that Tau$^{+/-}$ and Tau$^{-/-}$ mice are resistant to epileptiform activity and seizures induced by excitotoxic agents (*Roberson et al., 2007*; *Ittner et al., 2010*; *Roberson et al., 2011*). Tau knockdown using ASOs also has excitoprotective effects against hyperexcitability in mice (*DeVos et al., 2013*), complemented by our finding of excitoprotective effects against BIN1-induced hyperexcitability (*Figure 6*).

The precise effects of AD-associated *BIN1* variants remains to be fully understood, but their effects are likely to be mediated through changes in expression levels since they do not affect the coding sequence. For example, the risk allele of the AD-associated rs744373 variant drives increased expression of BIN1 (*Bungenberg et al., 2016*). While another early report suggested increased BIN1 in AD (*Chapuis et al., 2013*), subsequent reports suggest that variants may reduce BIN1 expression

(*Glennon et al., 2013*; *Holler et al., 2014*), and the effects may also differ between the neuronal and ubiquitous isoforms (*De Rossi et al., 2016*). Ongoing studies will provide additional evidence about the directionality of *BIN1* variant effects on expression, but our findings are consistent with either increases or decreases in BIN1 contributing to network hyperexcitability in AD, since we found that higher BIN1 was associated with higher activity as a general effect in both excitatory and inhibitory neurons. That is, either increased BIN1 in excitatory neurons with corresponding increased excitatory activity, or reduced BIN1 in inhibitory neurons with corresponding reduced inhibition, could lead to network hyperexcitability. Further study will be required to better understand both the effects of AD-associated *BIN1* variants and the relative balance between excitatory and inhibitory neuron effects of BIN1. In addition, BIN1 expression changes in oligodendrocytes or microglia also warrant study for their potential roles in AD (*De Rossi et al., 2016*; *Nott et al., 2019*).

Our findings highlight the potential importance of Tau interactions with SH3 domain–containing proteins. We recently demonstrated that inhibiting Tau-SH3 interactions can reduce Aβ toxicity (*Rush et al., 2020*), and it is notable that BIN1 joins a growing list of SH3 domain–containing proteins that interact with Tau and are implicated in AD. Tau may act as a scaffolding protein through BIN1 interactions mediating membrane localization (*Figure 5A*), promoting network hyperexcitability through its SH3-domain containing binding partners. This could be a critical role for Tau and explain how mislocalization of Tau in AD contributes to the increased network excitability seen in AD pathogenesis. This would be consistent with the finding that reducing endogenous Tau prevents network hyperexcitability and Aβ-induced dysfunction in AD models.

In summary, we have shown that BIN1 promotes neuronal firing in a Tau-dependent manner. These data contribute new insights into the neuronal functions of BIN1, with implications for our understanding of AD.

# Materials and methods

**Key resources table**

| Reagent type (species) or resource | Designation | Source or reference | Identifiers | Additional information |
|---|---|---|---|---|
| Gene (*Homo sapiens*) | *BIN1* | NCBI | Gene ID 274 | |
| Antibody | Anti-NeuN Rabbit polyclonal | abcam | Cat# ab104225; RRID:AB_10711153 | ICC (1:500), Lot #GR3321966-1 |
| Antibody | Anti-GAD67 Mouse monoclonal | Millipore Sigma | Cat# MAB5406; RRID:AB_2278725 | ICC (1:500), Lot #3015328 |
| Antibody | Anti-BIN1 Rabbit polyclonal | Santa Cruz | Cat# sc-30099; RRID:AB_2243399 | ICC/PLA/IP (1:500), Lot #K1605; H-100 |
| Antibody | Anti-LVGCC-β1 Mouse monoclonal | abcam | Cat# ab85020; RRID:AB_1861569 | ICC/PLA/IP (1:1000), Lot #413-8RR-52 |
| Antibody | Anti-Tau Rabbit polyclonal | DAKO | Cat# A0024; RRID:AB_10013724 | ICC/IP (1:1000), Lot #20031827 |
| Antibody | Anti-BIN1 Mouse monoclonal | Santa Cruz | Cat# sc-13575; RRID:AB_626753 | ICC/IP (1:1000), Lot #L3014; 99D |
| Cell line (*Rattus norvegicus*) | Primary neuron | Charles River | | Fresh from E19 albino Sprague Dawley rats |
| Genetic reagent | AAV-BIN1-mKate2 | UPenn Vector Core | | AAV2 |
| Genetic reagent | AAV-mKate2 | UPenn Vector Core | | AAV2 |

*Continued on next page*

*Continued*

| Reagent type (species) or resource | Designation | Source or reference | Identifiers | Additional information |
|---|---|---|---|---|
| Sequence-based reagent | Tau ASO | PMID:23904623 IDT | | 5-ATCACTGATTTTGAAGTCCC-3 |
| Sequence-based reagent | Scrambled ASO | PMID:23904623 IDT | | 5-CCTTCCCTGAAGGTTCCTCC-3 |
| Commercial assay, kit | Duolink PLA kit | Millipore Sigma | Cat#s DUO92014; DUO92002; DUO92004 | |
| Transfected construct | GCaMP6f | Addgene | RRID:Addgene_40755 | |
| Transfected construct | mKate2 | PMID:25156556 Evrogen | Cat# FP184 | Actin was removed from the construct obtained |
| Transfected construct (*Homo sapiens*) | BIN1 | Horizon Discovery ORFeome Collaboration Clones | OHS5894-202501160 | Isoform 1 |
| Cell line (*Cricetulus griseus*) | CHO-K1 | Millipore Sigma | Cat# 85051005-1VL | Chinese Hamster Ovary cell line |
| Transfected construct | mKate2-Tau-mKate2 | PMID:25156556 | | |
| Transfected construct | Fyn-SH3-CBG | PMID:25156556 | | BIN1-SH3 or LVGCC-β1-SH3 was cloned in replacing Fyn-SH3 |
| Transfected construct (*Homo sapiens*) | BIN1-SH3 | IDT | AAC28646.1 | Codon optimized |
| Transfected construct (*Homo sapiens*) | LVGCC-β1-SH3 | IDT | AAA35632.1 | Codon optimized |

## Primary neuron cultures

Primary hippocampal culture protocols were adapted from *Rush et al., 2020*. Briefly, hippocampal tissue from E19 Sprague Dawley albino rat (*Rattus norvegicus*) embryos was harvested on ice in 4°C Hibernate E (Life Technologies, A1247601) and digested with 20 units/mL papain (Worthington Bio-chemical Corporation, LK003178) for 10 min at 37°C. Neurons were then dissociated by manual tritu-ration to a single-cell suspension in Neurobasal medium (Life Technologies, 21103049) supplemented with 1x B-27 (Gibco, 17504044), 2 mM L-Glutamine (Life Technologies, 25030081) and 10% premium select fetal bovine serum (Atlanta Biologicals, S11550). Neuronal plating condi-tions depended on the experiment, as follows.

### Multi electrode array cultures

For 6-well multielectrode array recordings, neurons were plated at 100,000 per well in six-well MEA plates (ALA Scientific, ALAMEA-MEMMR5). For 48-well plate multielectrode array recordings, neu-rons were plated at 30,000 per well in 48-well MEA plates (Axion Biosystems, M768-tMEA-48B-5).

### Calcium imaging, electrophysiology, and immunocytochemistry cultures

Neurons were plated at 50,000 neurons per well on 12 mm coverslips (Carolina Biological, 633029) in 24-well plates coated overnight at 4°C with 0.1 mg/mL Poly-D-Lysine (Sigma, P6407−10 × 5 MG) and 0.2 mg/mL laminin (Sigma, L2020-1MG) 24–48 hr prior to the neuron harvest, with the outer wells containing autoclaved ultrapure water (MilliQ filtered) to prevent evaporation.

## Immunoblotting

Neurons were plated at 200,000 per well in six-well plates (Corning, 08-772-1B) and maintained in a 37°C humidified incubator with 5% $CO_2$. 24 hr after plating, 75% of the medium was exchanged for serum-free Neurobasal supplemented with B-27 and L-Glutamine, with 5 µM cytosine β-D-arabino-furanoside (AraC, Sigma Aldrich, C6645) to inhibit glial proliferation. 50% medium changes were performed weekly with Neurobasal supplemented with B-27 and L-Glutamine until experiments were started at DIV 19–21.

## BIN1 constructs and vectors

A BIN1-mKate2 (GE Dharmacon, OHS5894-202501160) construct was developed to encode human BIN1 isoform 1 (593 AA, the major neuronal isoform) tagged with mKate2 (Evrogen, FP184, to allow for fluorescent visualization) at the C-terminus to allow for proper function of the N-terminal membrane-interacting BAR domain. A similar construct lacking the BAR domain (amino acids 32–273, BIN1-ΔBAR-mKate2) was produced as a BIN1 BAR domain deletion mutant. A construct encoding mKate2 only was used as a control. These constructs were then cloned into the CIGW vector (rAAV9-CBA-IRES-GFP-WPRE-rBG) (*St Martin et al., 2007*). Due to size limitations for efficient gene expression, the IRES-GFP was removed from the CIGW vector.

## Neuronal transduction

BIN1-mKate2 and mKate2 vectors were packaged into rAAV2 at the University of Pennsylvania Vector Core (stock titers: AAV-BIN1-mKate2: 3.2e12 genomes/ml, AAV-mKate2: 9.69e12 genomes/ml; used titers: AAV-BIN1-mKate2 and AAV-mKate2: 1e10 genomes/ml, used MOI: AAV-BIN1-mKate2 and AAV-mKate2: 200,000). AAV vectors were used in MEA and electrophysiology experiments. Neuronal cultures were transduced on DIV 2.

## Neuronal transfection

BIN1-mKate2, BIN1-ΔBAR-mKate2, and mKate2 vectors were used in transient transfections in calcium imaging experiments. Transfections were performed at DIV 14 using a calcium phosphate precipitation protocol adapted from *Frandemiche et al., 2014*. Briefly, Neurobasal medium was removed and kept until the last step of transfection used as conditioned Neurobasal medium. Neurons were washed for 1–1.5 hr in DMKY buffer containing 1 mM kynurenic acid, 0.9 mM NaOH, 0.5 mM HEPES, 10 $MgCl_2$, plus phenol red 0.05%, pH 7.4. Then, 3.5 µg of the vectors were mixed with 120 mM $CaCl_2$ in HBSS (Life Technologies, 14175095) containing 25 mM HEPES, 140 mM NaCl, and 0.750 mM $Na_2HPO_4$, pH 7.06, left for 20 min to precipitate the DNA, and applied to the primary hippocampal cultures for 30 min. The medium was then replaced with conditioned Neurobasal medium (Life Technologies, 21103049) and cultures were returned to the incubator until use.

## Multielectrode array recordings

### Multi Channel Systems MEA

MEA recording protocols were adapted from *Savell et al., 2019b*. Briefly, E19 rat hippocampal neurons were seeded to six-well MEAs containing nine extracellular recording electrodes and a ground electrode. Neurons were transduced with AAV expressing BIN1 or control constructs on DIV2. Transduced neurons had 50% medium changes with BrainPhys (StemcellTech, 05793) medium supplemented with N2A and SM1 at DIV 5 and 9 to promote maturation, then with supplemented Neurobasal (Life Technologies, 21103049) at DIV 12. 20 min MEA recordings were performed at DIV 12–13 in the temperature-controlled headstage at 37°C. Neuronal firing was amplified and acquired at 30 kHz, digitized, and further analyzed in MC_Rack (Multi Channel Systems). Data were filtered at 10 Hz and 10,000 Hz filters and thresholded to detect action potentials at each electrode. Detected action potentials were transferred to Offline Sorter (v. 4.0 Plexon) to differentiate multiple neurons detected with a single electrode using principal component analysis (PCA) of waveform properties. Offline Sorter automatically completes and plots PCA on waveforms for each electrode. Manual inspection of PCA, shape, inter-spike intervals, auto-correlograms, and cross-correlograms allowed us to distinguish between multiple units on a single electrode and to do per-neuron analyses. After waveforms were split into units, analysis of each unit's action potential frequency and burst firing was completed in NeuroExplorer (v. 5.0, Plexon) using the built-in Burst Analysis function, with

Poisson burst surprise = 5. Next, firing rates and bursting analysis were performed in NeuroExplorer (v. 5.0 Plexon). Researchers were blinded to experimental conditions performed in all MEA analyses.

## Axion Biosciences MEA

Single neuron electrophysiological activity was recorded using an Axion Maestero recording system as in *Savell et al., 2019a*. Briefly, neurons were plated on the 48-well MEA (Axion Biosystems, M768-tMEA-48B-5) with 16 extracellular recording electrodes and a ground electrode per well at a density of 30,000 neurons per well in Neurobasal medium (5 µL) with 10% FBS (Atlanta Biologicals, S11550) and placed in a 37°C incubator with 5% CO$_2$. After allowing neurons to attach to the plate for 2 hr, 300 µL serum-free Neurobasal (Life Technologies, 21103049) was added. The next day, AraC was added as with other experiments and a 50% medium change with BrainPhys (Stemcell Technologies Inc, 05790) supplemented with SM1 and L-glutamine was done at DIV 5. At DIV 6, neurons were treated with ASO to reduce Tau protein levels. At DIV 9, a 50% medium change was completed with supplemented BrainPhys, followed by a 50% medium change with supplemented Neurobasal at DIV 12. At DIV 13, neurons were recorded using Axion AxIS software for 15 min. Electrical activity was measured by an interface board at 12.5 kHz, digitized, and transmitted to an external computer for data acquisition and analysis in Axion AxIS Navigator software (Axion Biosystems). All data were filtered using dual 0.01 Hz (high pass) and 5,000 Hz (low-pass) Butterworth filters. Action potential thresholds were set automatically using an adaptive threshold for each electrode (>6 standard deviations from the electrode's mean signal). Neuronal waveforms collected in Axion AxIS Navigator were exported to Offline Sorter (v. 4.0 Plexon). Offline Sorter automatically completes and plots PCA on waveforms for each electrode. Manual inspection of PCA, shape, inter-spike intervals, auto-correlograms, and cross-correlograms allowed us to distinguish between multiple units on a single electrode and to do per-neuron analyses. After waveforms were split into units, analysis of each unit's action potential frequency and burst firing was completed in NeuroExplorer (v. 5.0, Plexon) using the built-in Burst Analysis function, with Poisson burst surprise = 5. Next, firing rates and bursting analysis were performed in NeuroExplorer (v. 5.0 Plexon). Researchers were blinded to experimental conditions performed in all MEA analyses.

## Antisense oligonucleotide application

Tau anti-sense oligonucleotide (ASO) sequences were adapted from *DeVos et al., 2013* and produced by Integrated DNA Technology (Tau ASO: 5-ATCACTGATTTTGAAGTCCC-3, Nontargeting control ASO: 5-CCTTCCCTGAAGGTTCCTCC-3). ASOs were dissolved to 100 µM in 10 mM Tris with 0.1 mM EDTA and stored at −20°C until use. At DIV 6, one week before testing for both MEA experiments and PLA, neurons were treated with ASO to a final concentration of 1 µM.

## Calcium imaging

Calcium imaging was adapted from *Léveillé et al., 2008*. Briefly, rat primary hippocampal neurons (DIV 14) were transfected (see Neuronal transfection section) with the genetically engineered calcium sensor GCaMP6f (gift from Dr. Alain Buisson, originally developed by Douglas Kim and GENIE Project, Addgene plasmid #40755, *Chen et al., 2013*). At DIV 21, the neurons were incubated for 15 min at room temperature in HEPES and bicarbonate buffered saline solution (HBBSS) containing 116 mM NaCl, 5.4 mM KCl, 1.8 mM CaCl$_2$, 0.8 mM MgSO$_4$, 1.3 mM NaH$_2$PO$_4$, 12 mM HEPES, 5.5 mM glucose, 25 mM bicarbonate and 10 µM glycine at pH 7.45. Neurons that were transfected with mKate2 or BIN1-mKate2 vectors (see BIN1 constructs and vectors section) were recorded for 8 min. Experiments were performed at room temperature with continuous perfusion at 2 ml/min with a peristaltic pump, on the stage of a Nikon A1R Confocal (Nikon, TE2000) inverted microscope equipped with a 100 W mercury lamp and oil-immersion Nikon 40x objective with 1.3 numerical aperture (Nikon, Tokyo, Japan). GCaMP6f (excitation: 340/380 nm, emission: 510 nm) ratio images were acquired at 8 Hz with a digital camera (Princeton Instruments, Trenton, NJ) using Metafluor 6.3 software (Universal Imaging Corporation, West Chester, PA, USA). Fluorescence ratios (340/380 nm) were converted to intracellular Ca$^{2+}$ concentration using the following formula:

$$[Ca^{2+}]_i = K_d \left( \frac{R - R_{min}}{R_{max} - R} \right) \left( \frac{F_0}{F_s} \right)$$

where $R$ is the measured ratio of 340/380 fluorescence, $R_{min}$ is the ratio measured in a $Ca^{2+}$-free solution, $R_{max}$ is the ratio measured in a saturated $Ca^{2+}$ solution, $K_d$ = 135 nM (the dissociation constant for GCaMP6f), and $F_0$ and $F_s$ are the fluorescence intensities measured at 380 nm, respectively, in a $Ca^{2+}$-free solution or in a saturated $Ca^{2+}$ solution.

## Electrophysiology

All electrophysiological recordings were performed in primary hippocampal neuronal cultures after 19–21 DIV. Whole-cell patch-clamp recordings were made from visually identified pyramidal neurons. Recorded signals were amplified with a MultiClamp 700B amplifier (Molecular Devices), filtered at 5 kHz, and sampled at 10 kHz with Digidata 1550A (Molecular Devices). Recordings were acquired using pClamp (v.10) and analyzed using Clampfit (Molecular Devices). Patch pipettes had a resistance of 2.5–5 MΩ when filled with the internal solution required for the experiments described below. All recordings were performed at room temperature (21–23˚C). Internal solution included 120 mM Cs-gluconate, 0.6 mM EGTA, 2.8 mM NaCl, 5 mM MgCl$_2$, 2 mM ATP, 0.3 mM GTP, 20 mM HEPES, and 5.0 mM QX-314. External solution included 119 mM NaCl, 2.5 mM KCl, 1.3 mM MgSO$_4$, 2.5 mM CaCl$_2$, 1 mM NaH$_2$PO$_4$, 26 mM NaHCO$_3$, 11 mM glucose (pH 7.3). Voltage-clamp recordings to measure sEPSCs were performed from cultured neurons by whole-cell patch-clamp holding the neurons at –70 mV with 100 µM picrotoxin (GABA$_A$R antagonist, Tocris, 11–281 G) in the bath solution. Voltage-clamp recordings to measure sIPSCs were performed from cultured neurons by whole-cell patch-clamp holding the neurons at 0 mV in 10 µM DNQX (AMPAR antagonist, Sigma, D0540-25MG), 100 µM APV (NMDAR antagonist, Tocris, 01-055-0), and 10 µM nifedipine (L-type VGCC antagonist, Sigma, N7634-25G) to enrich for sIPSCs from spontaneously active interneurons rather than from interneuron-driven by excitatory transmission.

## Immunocytochemistry (ICC) and NeuN quantification

ICC was adapted from *Rush et al., 2020*. Briefly, primary neurons on coverslips were fixed with 4% PFA and 4% sucrose in 1x PBS. Coverslips were permeabilized with 0.25% Triton X-100 in 1x PBS for 10 min at room temperature then blocked for one hour in 5% FBS in 1x PBS. Primary antibody for NeuN (abcam, ab104225, 1:500), GAD67 (Millipore Sigma, MAB5406, 1:500), BIN1 (Santa Cruz, sc-30099, 1:500), or LVGCC-β1 (Abcam, S7-18, 1:1,000) in 1% FBS in 1x PBS was applied overnight at 4˚C. Coverslips were then washed 3 × 5 min in 1x PBS, then incubated in Alexa Fluor fluorescent antibodies (1:1,000) in 1% FBS in 1x PBS for 1 hr at room temperature. Coverslips were washed 3 × 5 min in 1x PBS, then mounted in Prolong Diamond. For neuron quantification, 10 × 10 images at 20x were taken with an epiflourescent microscope and automatically stitched together using Nikon NIS-Elements. NeuN images were thresholded in ImageJ, then quantified using ImageJ (v. 2.0.0-rc-69/1.52 p) particle analyzer.

## Proximity Ligation Assay (PLA)

PLA was adapted from *Rush et al., 2020*. Briefly, neurons on coverslips were fixed and permeabilized as with ICC, then were incubated overnight with primary antibodies for BIN1 (Santa Cruz, sc-30099, 1:500) and LVGCC-β1 (Abcam, S7-18, 1:1,000) overnight at 4˚C, then PLA was performed using the Duolink In Situ Fluorescence kit (Sigma, DUO92004-100RXN). After PLA, coverslips were incubated with secondary antibody to view BIN1 and mounted with Duolink In Situ Mounting Medium with DAPI. Fluorescent images were taken using an epifluorescence microscope at 60x with four channels: DAPI (nuclei), FITC (PLA), and TRITC (mKate2). 7–9 images per slide were obtained and analyzed using ImageJ (v. 2.0.0-rc-69/1.52 p). PLA puncta were quantified using ImageJ particle analyzer, and the average number of puncta per field of view (FOV) or each coverslip was used for analysis.

## Bioluminescence Resonance Energy Transfer (BRET)

BRET was conducted as described in *Cochran et al., 2014*. Codon-optimized human BIN1-SH3 or LVGCC-β1-SH3 domains were fused on the C-terminus to click beetle green (CBG) luciferase (Promega, E1461) replacing the Fyn-SH3 in the previously described donor construct. Tau tagged at each terminus with mKate2 (Evrogen, FP184) served as the acceptor. Chinese hamster ovary (CHO) cells (Sigma, 85051005-1VL) obtained from ECACC (Lot number 12G006) were authenticated using DNA

Fingerprinting and DNA bar-coding sequencing and tested negative for mycoplasma contamination using PCR, a Vero indicator cell line, and Hoechst 33258 fluorescent detection system (Certificate of Analysis test number 47856). CHO cells were plated in 24-well opaque white plates (Promega, 6005168) using the manufacturer's instructions and transfected with donor and acceptor constructs using Fugene. Forty-eight hours later, fluorescence was read by excitation with a 530/25 nm filter and emission with a 645/40 nm filter on a Synergy2 (BioTek) to control for the concentration of the donor. Immediately after fluorescence measurement, D-luciferin (Promega, E1605) was added to a final concentration of 200 µM to each well. Two to 4 hr later, after the signal had stabilized, plates were read with 645/40 nm filter. Measured BRET fluorescence was normalized to mKate2 fluorescence.

## Co-immunoprecipitation

Mouse hemibrains were finely chopped while frozen, then thawed on ice in PBS plus protease inhibitors (Fisher PI-78439), phosphatase inhibitors (Sigma-Aldrich, P5726), and 1 mM of the cell-permeable cross-linker DSP (Fisher, PI-22585). Hemibrains were then homogenized for 15 s using a handheld Kontes Pellet Pestle homogenizer, then pipetted up and down 20x to obtain a smooth lysate. Lysates were spun $2 \times 10$ min at 800 x $g$, then cleared lysates were incubated for 15 min at 4°C on an end-over-end rotator. Next, lysates were brought to 100 mM Tris to inactivate DSP and incubated for another 15 min at 4°C on an end-over-end rotator. Samples were then diluted 1:1 with a mild co-IP buffer: 10 mM Tris (pH 7.5), 10 mM NaCl, 3 mM MgCl$_2$, 1 mM EGTA, and 0.05% Nonidet P-40, a mild lysis buffer previously shown to be amenable to co-IP experiment (*Filiano et al., 2008*). At this point, an input fraction was set aside before adding IP antibody to the lysate, with 5 µg of antibody used in each case. Lysate/antibody mixtures were incubated overnight on an end-over-end rotator. Next, 50 uL of Protein G–coated magnetic beads (Life Technologies, 10004D) were added and incubated for 8 hr at 4°C on an end-over-end rotator. Next, non-interacting lysate was removed, the bead/antibody/antigen complex was washed, then protein was eluted with 50 mM Glycine (pH 2.8) and neutralized with 1 M Tris, reduced with β-Mercaptoethanol and an 80°C incubation for 10 min, then cooled and probed by immunoblotting.

## Immunoblots

5 µg of immunoprecipitated samples were loaded and separated on 4–12% NuPage acrylamide gels (Invitrogen) with NuPage MOPS running buffer for 2 hr at 110 V. Next, proteins were transferred to Immobilon-FL PVDF membranes (Millipore) using the NuPage transfer buffer transfer system (Invitrogen) overnight. The membrane was blocked in LI-COR Odyssey blocking buffer (LI-COR, 927–40000) for 1 hr at room temperature and incubated with the appropriate primary antibody. After primary antibody treatment, membranes were washed three times in tris-buffered saline with 0.1% Tween (TBS-T), followed by incubation for 1 hr with Alexa Fluor 700– or 800–conjugated goat antibodies specific for mouse immunoglobulin G (1:20,000, LI-COR). Membranes were then washed three times in TBST-T, followed by a single wash in TBS, imaged on the LI-COR Odyssey fluorescence imaging system, and quantified using LI-COR Image Studio (v. 5.2.5).

## Animals

All breeding and experimental procedures were approved by the University of Alabama at Birmingham Institutional Animal Care and Use Committee and follow the guidelines by the National Institutes of Health. Male and female Tau$^{+/-}$ mice lacking exon 1 of MAPT gene were bred to obtain Tau$^{-/-}$ mice with littermate Tau$^{+/+}$ controls. Mice were maintained under standard laboratory conditions (12 hr light/dark cycle, 50% humidity, Harlan 2916 diet, and water ad libitum). Genotype was verified by standard PCR protocol.

## Statistics

Statistical distribution of data varied widely between data sets in this study, so we analyzed each data set for normality and analyzed using either parametric or non-parametric tests accordingly. The specific test used is indicated in the figure legend in each case. All statistical tests were performed with Prism 8 (GraphPad, v. 8.4.0).

## Acknowledgements

We thank all members of the Roberson lab for helpful discussions and critiques, and Andy West, David Standaert, and Alain Buisson for plasmids. This work was supported by the National Institutes of Health grants RF1AG059405, R01NS075487, R01MH114990, T32NS095775, and T32NS061788, the Alzheimer's Association, and the Weston Brain Institute. The authors declare no financial interests. EDR is an owner of intellectual property relating to Tau.

## Additional information

### Competing interests

Erik D Roberson: EDR is an owner of intellectual property relating to Tau. The other authors declare that no competing interests exist.

### Funding

| Funder | Grant reference number | Author |
|---|---|---|
| National Institutes of Health | RF1AG059405 | Erik D Roberson |
| National Institutes of Health | R01NS075487 | Erik D Roberson |
| National Institutes of Health | R01MH114990 | Jeremy J Day |
| National Institutes of Health | T32NS095775 | Yuliya Voskobiynyk |
| National Institutes of Health | T32NS061788 | Jonathan R Roth |
| Alzheimer's Association | | Erik D Roberson |
| Weston Brain Institute | | Jonathan R Roth<br>Travis Rush<br>Erik D Roberson |

The funders had no role in study design, data collection and interpretation, or the decision to submit the work for publication.

### Author contributions

Yuliya Voskobiynyk, Conceptualization, Resources, Data curation, Software, Formal analysis, Supervision, Funding acquisition, Validation, Investigation, Visualization, Methodology, Writing - original draft, Project administration, Writing - review and editing; Jonathan R Roth, Data curation, Software, Funding acquisition, Validation, Investigation, Visualization, Methodology, Project administration, Writing - review and editing; J Nicholas Cochran, Travis Rush, Data curation, Software, Formal analysis, Validation, Visualization, Methodology, Writing - review and editing; Nancy VN Carullo, Data curation, Software, Formal analysis, Supervision, Methodology, Writing - review and editing; Jacob S Mesina, Mohammad Waqas, Rachael M Vollmer, Data curation, Formal analysis, Writing - review and editing; Jeremy J Day, Resources, Software, Supervision, Funding acquisition, Writing - review and editing; Lori L McMahon, Methodology, Writing - review and editing; Erik D Roberson, Conceptualization, Resources, Formal analysis, Supervision, Funding acquisition, Investigation, Methodology, Project administration, Writing - review and editing

### Author ORCIDs

Yuliya Voskobiynyk (iD) https://orcid.org/0000-0003-4169-3002
Jonathan R Roth (iD) https://orcid.org/0000-0001-8978-4507
Nancy VN Carullo (iD) http://orcid.org/0000-0001-9197-5046
Jeremy J Day (iD) http://orcid.org/0000-0002-7361-3399
Lori L McMahon (iD) http://orcid.org/0000-0003-1104-6584
Erik D Roberson (iD) https://orcid.org/0000-0002-1810-9763

## Ethics

Animal experimentation: This study was performed in strict accordance with the recommendations in the Guide for the Care and Use of Laboratory Animals of the National Institutes of Health. All of the animals were handled according to approved institutional animal care and use committee (IACUC) protocols (#20450) of the University of Alabama at Birmingham. The protocol was approved by the Committee on the Ethics of Animal Experiments of the University of Alabama at Birmingham.

## Decision letter and Author response

Decision letter https://doi.org/10.7554/eLife.57354.sa1
Author response https://doi.org/10.7554/eLife.57354.sa2

## Additional files

### Supplementary files

• Transparent reporting form

### Data availability

All data generated or analysed during this study are included in the manuscript and supporting files. Source data files have been provided for Figures 6; high throughput raw electrophysiologic recordings of neuronal activity using Axion Biosciences Maesto are deposited on Dryad at https://doi.org/10.5061/dryad.rbnzs7h8z. Brief Analysis used is described in the methods section; in-depth analysis description is publicly available at https://www.axionbiosystems.com/products/software/neural-module.

The following dataset was generated:

| Author(s) | Year | Dataset title | Dataset URL | Database and Identifier |
|---|---|---|---|---|
| Voskobiynyk Y, Roth JR, Cochran JN, Rush T, Carullo NVN, Mesina JS, Waqas M, Vollmer RM, Day JJ, McMahon LL, Roberson ED | 2020 | Data from: Alzheimer's disease risk gene *BIN1* induces Tau-dependent network hyperexcitability | https://doi.org/10.5061/dryad.rbnzs7h8z | Dryad Digital Repository, 10.5061/dryad.rbnzs7h8z |

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
