## [Decision Letter]

**Acceptance summary:**

Genetic studies clearly indicate that BIN1 modifies risk for developing Alzheimer's disease, but the mechanisms underlying this biology are poorly understood. In this study, Voskobiynyk et al. performed a rigorous set of studies that places BIN1 squarely in the tau-associated network hyperexcitability problem, with a mechanistic link via a direct interaction of BIN1 with L-type voltage-gated calcium channels (LVGCCs). This provides a much deeper understanding of not only normal BIN1 function, but how it may contribute to Alzheimer's disease pathogenesis.

**Decision letter after peer review:**

Thank you for submitting your article "Alzheimer's disease risk gene BIN1 induces Tau-dependent network hyperexcitability" for consideration by *eLife*. Your article has been reviewed by two peer reviewers, and the evaluation has been overseen by a Reviewing Editor and Huda Zoghbi as the Senior Editor. The following individuals involved in review of your submission have agreed to reveal their identity: Miranda Reed (Reviewer #1); Brian Kraemer (Reviewer #2).

The reviewers have discussed the reviews with one another and the Reviewing Editor has drafted this decision to help you prepare a revised submission.

Summary:

In this manuscript, Voskobiynyk et al. describe a new function for BIN1 related to network hyperexcitability that is directly modulated by the presence of tau. BIN1 is one of many Alzheimer's risk genes for which we have very little functional data. Using multielectrode arrays and calcium imaging in primary rat neurons, these authors demonstrate that human BIN1 expression induces network hyperexcitability and that these readouts are modulated by tau expression, likely via interactions with L-type voltage gated calcium channels. This data provides important functional understanding of how BIN1 influences Alzheimer's risk and sets the stage for future functional studies in animal models.

Essential revisions:

The invited reviewers and the reviewing editor unanimously agree that this study reveals important new insights into BIN1 function, and that the data were designed and carried out rigorously. A few additional points of clarification are needed that would greatly strengthen the manuscript.

1) Please include data regarding the excitatory/inhibitory balance for Figure 2 if available. Related to these conclusions, discussing the reversal potential how sIPSCs were measured at 0 mV is needed to better put these electrophysiological aspects in perspective.

2) It is also necessary to provide additional details on levels of BIN1 in DIV12 vs. DIV19 neurons (and state of neuronal health at both time points)

---

## [Author Response]

Essential revisions:The invited reviewers and the reviewing editor unanimously agree that this study reveals important new insights into BIN1 function, and that the data were designed and carried out rigorously. A few additional points of clarification are needed that would greatly strengthen the manuscript.1) Please include data regarding the excitatory/inhibitory balance for Figure 2 if available.

This is an interesting question that is a bit challenging to answer directly using dissociated neurons as we did in this study. The best and most direct way to measure excitatory/inhibitory (E/I) balance is by stimulating to evoke a monosynaptic EPSC followed by a disynaptic IPSC, as occurs in intact CA3-CA1 circuits in hippocampal slices, and then calculating the evoked E/I ratio (see, for example, (Stewart et al., 2020). Because dissociated hippocampal neuron cultures in vitro lack the anatomic pattern of synaptic connectivity in slices, it is not possible to do this in our system. Furthermore, recording sEPSCs and sIPSCs simultaneously in the same cell requires voltage clamping the neuron at a holding potential that allows sEPSCs to be electrically isolated from sIPSCs (i.e., at which EPSCs are inward currents and IPSCs are outward currents). Under our recording conditions, a patched neuron would have to be held at around –30 mV, a potential at which the driving force would be ~30 mV for both EPSCs and IPSCs, since sEPSC_rev_ ≈ 0mV and IPSC_rev_ ≈ –60 mV. We did not take this approach with our recordings due to concern that the small driving force would yield synaptic events too small to be easily resolved during analysis (see answer to next question).

An alternative approach to answering this question is to indirectly assess E/I balance mathematically by calculating the ratios of sEPSCs to sIPSCs frequency and amplitude (Author response table 1). This is less ideal than the direct approach, as the sEPSCs and sIPSCs were not recorded in the same cells so formal statistical comparisons are not possible. But consistent with our findings of BIN-induced network hyperexcitability, higher BIN1 levels increased the E/I ratio of both frequency (from 0.24 ± 0.71 to 0.47 ± 4.30) and amplitude (from 0.37 ± 0.27 to 0.43 ± 0.97).

**Author response table 1. resptable1:** Calculated E/I ratios of spontaneous postsynaptic current frequencies and amplitudes.

	mKate2 Frequency, mean ±SEM	BIN1-mKate2 Frequency, mean ±SEM	mKate2 Amplitude, mean ±SEM	BIN1-mKate2 Amplitude, mean ±SEM
sEPSCs	1.65 ± 26.59 Hz	3.53 ± 65.58 Hz	31.15 ± 0.40 pA	29.02 ± 0.53 pA
sIPSCs	6.82 ± 37.61 Hz	7.57 ± 15.25 Hz	83.07 ± 1.49 pA	67.73 ± 0.55 pA
E/I ratio	0.24 ± 0.71	0.47 ± 4.30	0.37 ± 0.27	0.43 ± 0.97

Related to these conclusions, discussing the reversal potential how sIPSCs were measured at 0 mV is needed to better put these electrophysiological aspects in perspective.

To assess whether higher BIN1 expression modulates sIPSCs, we performed whole-cell patch-clamp recordings from visually identified pyramidal neurons using the identical internal solution that we used for measuring sEPSCs (Cs-gluconate internal solution), which has a chloride reversal potential (E_Cl_^-^ = –60 mV) near the resting membrane potential as is typical for hippocampal neurons. Therefore, to ensure that the driving force was large enough for us to easily resolve sIPSCs, patched cells were voltage clamped at 0 mV and recorded in the presence of inhibitors of AMPARs (10 µM DNQX) and NMDARs (100 µM APV). Thus, sIPSCs were both pharmacologically and electrically isolated from sEPSCs because sEPSCs reverse at 0 mV. In addition, an L-type VGCC antagonist (10 µM nifedipine) was employed to prevent contamination from active currents. These recording conditions are often used to enrich for sIPSCs from spontaneously active interneurons rather than from interneurons whose excitation was driven by excitatory transmission (Salin and Prince, 1996; Kotak and Sanes, 2000, 2014).

2) It is also necessary to provide additional details on levels of BIN1 in DIV12 vs. DIV19 neurons (and state of neuronal health at both time points).

We agree that these are important points and have added additional data to Figure 1 to address them (new Figure 1A-C). We demonstrated in the original submission that there were no significant differences in neuronal survival at DIV12 in any of the groups (original Figure 1H, I, which is now Figure 1D, E). For the new data, we grew neurons on coverslips, transduced with AAVs as before, then fixed and performed ICC to measure BIN1 levels and to verify neuronal health by evaluating structural integrity with Tau staining. BIN1 levels were ~8-9 times higher in neurons with AAV-BIN1-mKate2 than in untransduced or AAV-mKate2 neurons, and BIN1 levels were consistent from DIV 15–27. Through at least DIV 27, well past our experimental timepoints, neurons were morphologically normal without any evidence of adverse effects, as demonstrated with Tau ICC.

As a functional assessment, we used whole-cell current clamp to record the resting membrane potential (RMP) and input resistance (R_in_) of neurons at DIV 19, which also indicated no detrimental effects of AAV-BIN1-mKate2 (Table 1). RMP and R_in_ were similar across three groups (mean ± SEM) and consistent with healthy cultured neurons (Meadows et al., 2016).

Altogether, these results demonstrate that BIN1 levels were consistently increased over time after transfection and that the neurons in all groups were healthy well past our experimental timepoints.

References:

Kotak VC, Sanes DH (2000) Long-Lasting Inhibitory Synaptic Depression is Age- and Calcium-Dependent. The Journal of Neuroscience 20:5820.Kotak VC, Sanes DH (2014) Developmental expression of inhibitory synaptic long-term potentiation in the lateral superior olive. Frontiers in neural circuits 8:67-67.Meadows JP, Guzman-Karlsson MC, Phillips S, Brown JA, Strange SK, Sweatt JD, Hablitz JJ (2016) Dynamic DNA methylation regulates neuronal intrinsic membrane excitability. Sci Signal 9:ra83.Salin PA, Prince DA (1996) Spontaneous GABAA receptor-mediated inhibitory currents in adult rat somatosensory cortex. J Neurophysiol 75:1573-1588.Stewart LT, Abiraman K, Chatham JC, McMahon LL (2020) Increased O-GlcNAcylation rapidly decreases GABA(A)R currents in hippocampus but depresses neuronal output. Sci Rep 10:7494.